# Consumer Preferences of Sustainability Labeled Cut Roses in Germany

**Daniel Berki-Kiss * and Klaus Menrad**

Chair of Marketing and Management of Biogenic Resources, University of Applied Sciences Weihenstephan-Triesdorf, TUM Campus Straubing, Petersgasse 18, 94315 Straubing, Germany; klaus.menrad@hswt.de
* Correspondence: daniel.berki-kiss@hswt.de

**Abstract:** The study investigated preferences of consumers of food retailing outlets in Germany for sustainability labeled cut roses. A sample of 1201 respondents of an online survey was used to analyze their preferences based on a choice-based conjoint experiment in which a bunch of 10 roses was considered which differed concerning the labeling certificate, country of production of the plants, price, packaging, smell, and blossom size of the roses. Latent class analysis revealed existence of consumer heterogeneity with around two thirds of the respondents being strongly in favor of sustainability labels. Thereby Fairtrade labeled roses got an overall positive assessment whereby organic roses were only preferred by 31% of the respondents. In addition, paper or no packaging, strong smell and uniform big blossom sizes got overall positive consumer evaluations in the experiment. The study concludes that sustainability labeled plants might be an option for producers to append additional value to horticultural products in Germany.

**Keywords:** sustainability labels; cut roses; consumer heterogeneity; choice-based conjoint experiment

## 1. Introduction

The notions sustainability and sustainable consumption have been increasingly used over the last decades and appear to have become inherent components of the private sector as well. Sustainability is defined by the United Nations as: "development that meets the needs of the present without compromising the ability of future generations to meet their own needs" [1] (p. 41). Sustainable consumption as a key factor therefore refers to consumers' sustainability related attitudes and is associated directly with consumer behavior, especially how an individual's actions influence present and future outcomes [2]. In order to emphasize sustainable consumption and to inform consumers about relevant sustainability criteria certified products are provided with corresponding labels.

Fair, organic, local, green or other forms of sustainable consumption are considered by a growing number of households as shown by recent studies from different countries dealing with various product groups [3,4]. A considerable proportion (47%) of EU citizens´ purchasing decisions is supposedly influenced by sustainability labels like organic-labeling [3]. Results from Tanner and Kast [5] suggest that positive attitudes toward environmental protection, fair trade, local production, and sufficient consumer knowledge promote green food consumption. Hanss and Böhm [6] also demonstrate that information about sustainability issues drive individuals to choose certified ecological products. The share of sustainability labeled products of the food retailer MIGROS that is operating in several European countries has increased by 6.6% in one year [7]. As reported by Grunert et al. [8] consumers are guided (with exception of food choices) by sustainability labels depending on the level of their motivation and understanding.

Besides sustainably produced food products, other agricultural and horticultural products had to be taken into account in order to fulfil the rising demand of retailers and consumers in this area. According to the statistics of the agricultural market information company (AMI) [9] the market of ornamental plants and flowers had annual sales of about €8 billion in Germany in the last decade. Thereby the market segment of cut flowers achieved sales of €3 billion in 2016 of which cut roses had a dominant share of 44.0% of sold cut flowers in Germany [10,11]. Based on a report of CBI [12], "Germany is the largest market for cut roses in Europe"; with an increase of imports from €272 million to €309 million between 2011 and 2015.

Similar to other product groups, designated labels aiming to inform consumers about specific sustainability aspects have been developed and introduced in the markets of agricultural and horticultural products in recent years [13]. The market of sustainably cultivated agricultural products in Germany started with certified organic herbs, fruits and vegetables when the DEMETER organization introduced their first label back in 1928 [14]. In the following decades and years, other aspects of sustainability (e.g., locally produced, peat-free, enhanced $CO_2$-footprint) have gained relevance in the field of ornamental plants as well. Just recently, two new consumer labels were launched on the ornamental plants market in Germany: the GGN label (GlobalGAP Number) in 2017 [15] and the Fair Flora-Label from FloraZON [16]. These labels stand for a guaranteed overall sustainable production thereby aiming to directly motivate consumer choices.

From producers´ point of view, sustainability labels are often accompanied by substantial auditing fees and lower premiums than expected. This in turn leads to adoption of inferior labels with less-stringent social and ecological requirements [17]. However, the effect of sustainability labels on consumers is controversially discussed in literature: on one hand, it is necessary to communicate product related sustainability attributes toward consumers and thereby support product differentiation, on the other hand the existing plethora of certificates results in a confusing assortment and overwhelmed consumers [18–20]. Some recent scientific studies related to sustainability in horticultural plants analyzed the consumer-related effects of "fair" labels which consider primarily social aspects particularly in developing countries. These studies found positive preferences and consumer reactions (e.g., higher willingness-to-pay) for "fair" labeled ornamental plants [21–23]. Besides that, little is known to which extent some sustainability criteria demanded by certifications on the production side are affecting consumer choices of horticultural products. Results from recent scientific studies show that the health aspect is influencing choices of purchase of sustainable products [24,25], but this aspect is considered less relevant in relation to non-edible products. In the case of horticultural plants, consumer segments have been identified by Behe et al. [26] in the USA and Canada, whose purchase decisions for ornamental plants were determined by ecological aspects, and consumer preferences of additional ecologically-friendly practices were analyzed by other authors in recent years [27–29]. Further, Gabriel [30] could also demonstrate for Germany that environmental aspects were the main motive for purchasing plants in biodegradable pots, despite a higher price compared to the standard product. Previous studies also identified region of origin as an important product attribute for consumers when purchasing food products [31], but also plants from domestic or regional production are preferred [21,32–34].

Given the fragmented insights gained in previous studies, the influence of labeling and some relevant sustainability criteria on consumer choice of ornamental plants needs further exploration. Thus, the paramount aim is to identify sustainability criteria, which support consumer choices related to the analyzed cut roses as an example of horticultural products. Thereby the main emphasis of this study is

- Identifying relevant sustainability criteria for consumers related to production and distribution of cut roses
- Analyzing consumer preferences for cut roses with different sets of attributes focusing on the supply in food retailing outlets in Germany
- Clarifying the existence of consumer heterogeneity related to sustainability labeled cut roses.

The study on hand specifically contributes to the scientific literature by examining the preference heterogeneity of consumers concerning sustainability characteristics of horticultural plants. Two well-known consumer labels Fairtrade and Organic were integrated in order to perform a direct comparison of the social and ecological dimensions of sustainability in the case of the analyzed cut roses. To avoid socially desired responses, the current market situation with respect to product attributes and prices of cut roses in Germany is simulated in a choice-based conjoint (CBC) experiment. However, we do not aim at carrying out another willingness-to-pay study as often seen before, but intend to identify drivers for choosing the sustainable over the conventional alternative of a non-edible mass-product sold in food retail stores. Hence, our study analyzes the relevance and estimations related to different dimensions of sustainability in the view of consumers, including the inherent quality of the analyzed product (i.e., cut roses) in this process, and aims to identify consumer heterogeneity not only within the entire target group but also related to the different dimensions of sustainability.

## 2. Materials and Methods

To analyze the preferred attributes of sustainability labeled roses, a choice-based conjoint analysis was conducted. Consumer segments could be determined by applying a latent class analysis afterwards.

### 2.1. General Characteristics of the Method

The aim of a conjoint analysis is to determine utility values for selected attributes of a product by means of rank ordering of orthogonal arrays [35–37]. In general, a choice-based conjoint analysis incorporates a decision by a participant of the same product which varies among the attributes. Participants are assumed to choose the alternative which has the highest utility for them, thereby implicitly making trade-offs between the attributes. Through changing the levels of each attribute for the products, their impact on the choice of products can be calculated.

Choice-based conjoint analysis is a widespread methodology in science and practice of market research for analyzing preferences due to its realistic approach which simulates real buying decisions and has been applied in several studies dealing with consumer preferences for environmentally friendly products [38,39]. Furthermore, hypothetical attributes can be integrated if new or rarely available products are analyzed, as it is the case for organic roses which are currently rarely on the market in Germany.

### 2.2. Implementation of the Choice-Based Conjoint Experiment

The development and application of a choice-based conjoint experiment includes various steps. After the defining question, relevant and realistic product attributes and characteristics have to be identified and an appropriate experimental design has to be implemented. Then the questionnaire is developed, and an appropriate sample is chosen. The statistical analysis of the results completes the process.

When conducting a choice-based conjoint analysis, it is advised to keep the number of relevant attributes low to avoid excessive demands on the respondents which could negatively influence the quality of the results [40,41]. In this study, four different attributes were used including the product price which was linked to another product attribute. These attributes were derived from scientific literature, an analysis of available products, the outcome of expert interviews, as well as a pre-study aiming to investigate relevant sustainability criteria for purchaser of horticultural products.

2.2.1. Expert Interviews

After screening scientific literature, advice was sought from actors related to the production of horticultural plants at an international fair (IPM 2017) in Essen, Germany. In total, six expert interviews were carried out with participants representing positions from seed producing companies, retailing and wholesale companies to a certification organization (Transfair e.V.). All of the invited experts were able to share valuable insights owing to their longstanding and successful careers and due to their

diverse fields of activities within the market of ornamental plants. The interview partners were chosen carefully from the exhibitors' list of the IPM 2017 after exploring their backgrounds. In total, 18 of them were contacted by email to schedule an appointment, but finally representatives for six companies accepted our invitation. Each interview lasted approximately 60 min, was recorded and transcribed later. The covered questions were related to personal experiences with aspects of sustainability within the own company, labeling, and the communication towards consumers out of perspective of producers and the certification organizations.

The expert interviews showed that sustainability is an important driver for the marketing and selling of ornamental plants. Thereby social aspects were regarded to have the greatest influence on consumers since it is easier for consumers to identify themselves with these aspects. In addition, ecological aspects like reduced water consumption, reduced plastic waste, reduction of peat-use, and protection of insects seem to affect the purchase choices of consumers. In addition, the interview partners also described the difficulties of forwarding information to the buying consumers due to e.g., limited space on the packaging, labels are not known by consumers or not easy to understand, also additional information at the point-of-sale is difficult to realize and often not used by consumers. According to experts´ opinions, economic aspects are less relevant for private consumers, partially due to their restricted price knowledge [42]. Nevertheless, the opinion that sustainability labels are affecting consumer choices positively was supported by the majority. The organic BIO-labels (ECO-labels) and Fairtrade were mentioned as the most common ones, which are specially designed for consumers and this tendency is also endorsed in a study of the Zukunftsinstitut [43] and a report from TransFair [44]. The interviewed experts agreed that sustainability labels are gaining momentum in the field of ornamental plants by raising awareness and guaranteeing a specific "quality" during the production process. Furthermore, most of the interviewed experts confirmed that regionally produced horticultural products might be an interesting option following the currently running trend in food consumption in Germany.

### 2.2.2. Pre-Study on Sustainability Criteria

Following the expert interviews a short online-based pre-study was carried out in order to more clearly define relevant sustainability criteria in horticultural plants. The survey was designed with the online tool LimeSurvey (version 2.67.3), distributed via local newspaper, email and social media platforms and no incentives were offered to the participants: a total of 143 people participated in this pre-study, which was conducted for two weeks in August 2017. After cleaning the data, answers of 99 participants were finally taken into consideration. Due to the high consumer demand and availability of roses all over the year, we decided to focus our pre-study on cut roses and participants were asked to evaluate a list of the potentially relevant purchasing criteria for cut roses using a 5-point Likert-scale from "very important" to "not important at all." The results of this question are shown in Figure 1 indicating that certifications especially the BIO label (7.1%) is considered to have a low impact on consumer choice. This result might be explained by the currently low availability of organic roses in the German market [45,46]. On the other hand, social aspects in particular no child labor (62.6%) is affecting the purchase decision strongly. Fair working conditions and the protection of farmers (38.4% each) also seem to have an impact on consumer choice although the corresponding Fairtrade label only was selected by 19.1% of the participants. Consumers also attain relevance to some inherent characteristics like the color of the roses (46.5%), a high quality (43.4%), and the form of the blossom (34.4%). In contrast to features of social sustainability, ecological aspects seem to have a lower impact on consumer choice. Besides reduction of plastic waste (44.4%), in third place all the other ecological aspects are ranked between 33.3% for protection of bees and 14.1% for reduced peat use (Figure 1).

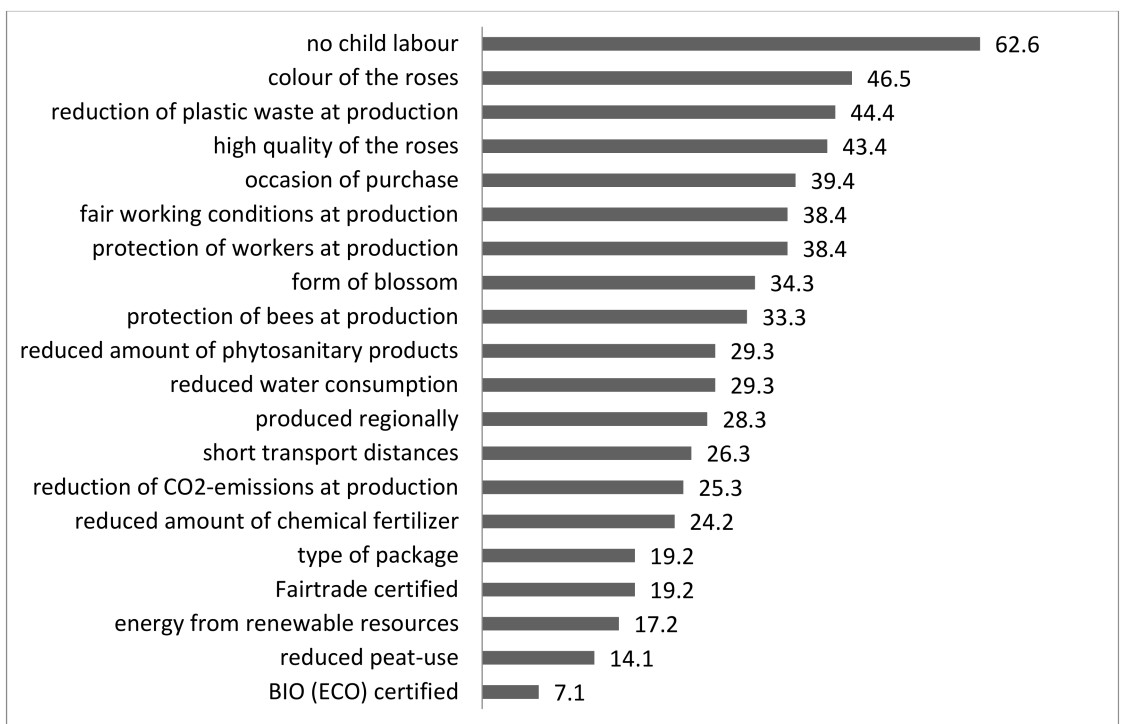

**Figure 1.** Relative importance of sustainability criteria when purchasing cut roses [47].

Further, the participants of the pre-study were asked about recognition and trust regarding four defined sustainability labels which are currently used on the German market. Three of the chosen labels are associated with ornamental plants (i.e., Fairtrade, BIO and Das Grüne Zertifikat) but also used in other segments (e.g., food products). These are the labels also mentioned by the experts in the interviews to be the most common for consumers. One additional label connected to environmental protection ("Blauer Engel") was included in the pre-study as well due to its long presence on the market. The results of this pre-study show, that Fairtrade (96.8%) and BIO (Organic) (60.6%) are the two most prominent labels in connection with ornamental plants. Although, when it comes to trust there is a noticeable difference between the scores of Fairtrade (84.5%) and BIO (39.3%). Since the selected labels are particularly indicators for social and ecological sustainability, we also asked if participants know the meaning of these terms. More than half of them tend to be aware of the meanings of the two concepts (65.0% for ecological sustainability and 58.0% for social sustainability).

*2.3. Selected Attributes for the Experiment*

The product attributes of the analyzed cut roses chosen for this CBC experiment are presented in Table 1. In order to keep the number of attributes used in the choice-based conjoint analysis comprehensibly low we focused on the findings of the pre-studies, especially the ranking presented in Figure 1. According to the experts´ statements, the labels Fairtrade and BIO are currently the two most relevant on the market hence we decided to include these two in the conjoint analysis despite their inferior rankings. Ultimately, the Fairtrade label represents the consumers´ most important aspects of social sustainability (i.e., no child labor, fair working conditions, and protection of workers at production). Similarly, the BIO-label was added, due to the strong tie to ecological sustainability attributes, expressing the consumers´ environmental consciousness like resource conservation in general. Besides sustainability aspects, inherent quality characteristics of the roses are also considered to be relevant, when reproducing a purchase situation.

**Table 1.** Product attributes of the choice-based conjoint experiment.

| Attribute Name | Attribute Level | Source |
|---|---|---|
| **Blossom size** | | |
| | Uniform big size blossoms; uniform small size blossoms; varying sizes of blossoms | Analysis of available products; expert interviews; pre-study on sustainability criteria; Prince et al. 1980 |
| **Scent** | | |
| | Strong scent; weak scent; no scent | Analysis of available products; expert interviews; pre-study on sustainability criteria; Doucé et al. 2013; Ellen and Bone 1998 |
| **Type of packaging** | | |
| | Paper packaging; plastic packaging; no packaging | Analysis of available products; expert interviews; pre-study on sustainability criteria; Steenis et al. 2017; Magnier et al. 2016; Lindh et al. 2016a; Lindh et al. 2016b |
| **Labeling /COO/price** | | |
| | Fairtrade/Kenya/support of developing countries/€1.99; BIO/Germany/support of regional production/€3.99; No label/no additional information/€1.65 | Analysis of available products; expert interviews; pre-study on sustainability criteria; Michaud et al. 2013; Klähre 2016; Hudson and Griffin 2004; Yue et al. 2011a,b |

The specific product attributes which had no reference to sustainability included the size of the blossom as well as the scent of the roses as quality characteristics inherent to these plants. Although appearance of horticultural plants was identified as a relevant quality parameter of consumers in previous studies (e.g., [34,48,49] most of these studies concentrated on the role of color of the blossom or characteristics of leaves. In an analysis of products already available on the market, the results of the performed expert interviews as well as the results of a pre-study on sustainability criteria showed the relevance of the size of the blossom as well as the smell as important attributes for consumers when buying roses. Both attributes are difficult to assess for consumers in quantitative terms so that they were transferred to three qualitative levels (Table 1) for each attribute.

In order to complement the environmental aspects of the analyzed product, we added the type of packaging of the roses as a separate attribute to the conjoint experiment. According to previous studies [50–52], consumer associations regard packaging mostly linked to convenience and sustainability. These findings also demonstrated that further benefits like e.g., naturalness, healthiness and overall quality of the packed product are related with the type of packaging. Lindh et al. [50] pointed out that paper-based packaging is considered more environmentally friendly than plastic. For defining the different attribute levels, we compared the currently used types of packaging for cut roses in the German market, which resulted in the three levels plastic, paper, and no packaging. During the expert interviews and in the conducted pre-study, especially the reduction of plastic waste was mentioned to have an important impact on consumer choices in the horticultural field.

In addition to the attribute "packaging," the other sustainability related aspects were combined in a specific attribute which was called "labeling/COO/price." One part of this attribute relates to the specific conditions during production of roses. Since consumers cannot control or evaluate these

conditions even after the product is bought or consumed, such characteristics are regarded as credence attributes [23,53]. The specific conditions during the production of credence products are generally transferred to consumers using specifically defined certification schemes as well as the respective labels to inform them. This was the reason why we named the respective attribute "labeling/COO/price." All of the sources for this attribute listed in Table 1 support the idea that the sustainability certifications Fairtrade and Organic are the most relevant consumer labels in Germany in this field. Therefore, we included these attribute levels with their original labels in the CBC experiment complemented with a "No label"-alternative (Figure 2). Since currently the label of the Organic farmers' organization "Bioland" is mostly used in the market of roses in Germany, this particular label was included in the experiment.

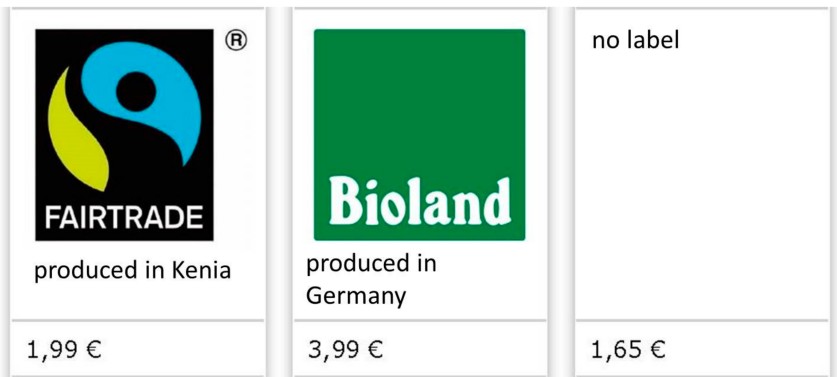

**Figure 2.** Illustration of the alternatives in the CBC for the combined attribute "labeling/COO/price."

In order to avoid unrealistic product alternatives, we combined the labels with two other factors. The origin of the plants was identified as an important selection criterion of consumers in previous studies [34,53]. Thus, we linked a specific country of origin (COO) to the three levels of this attribute. Since most of the Fairtrade roses sold in Germany are imported from Africa [54], Kenya as the most important production country of roses [55] was used for the "Fairtrade" level by adding the hint "supporting developing countries" (Table 1). Currently, the few organic roses that are available on the market in Germany [56] are produced domestically. Consequently, production of organic roses in a different country than Germany had to be excluded in the CBC experiment which is what we realized with emphasizing "support of regional production" (Table 1). The "No label"-alternative had no information about the country of origin in order to mimic the current situation on the market of roses in Germany.

Since the prices of products are essential in any economic transaction and were found to strongly impact consumers' decisions, prices are mostly included in studies analyzing consumer preferences [23,33]. In order to collect real-life market prices, we checked the prices of roses at local providers with respect to the specific labels and used a constant average price for each label considering a bundle of 10 roses. Since we did not intend to analyze the specific willingness-to-pay for the different attribute levels, but to focus on the preferences for sustainability criteria as they emerge currently on the market of roses in Germany, the function of the price in our CBC experiment was limited to imitate a rather realistic shopping situation. Therefore, we used constant prices for the three "labeling" levels with the price for organic roses produced in Germany being the highest at €3.99/10 pieces, followed by Fairtrade roses produced in Kenya for €1.99/10 pieces, and the alternative without any labeling was offered for the lowest price of €1.65/10 pieces.

Altogether, the attribute "labeling/COO/price" examines the combined effect of the three interrelated aspects specific conditions during production, country of origin and price. The argument behind this combined attribute is that the analyzed sustainability certifications go hand in hand with a given country of origin and a relatively stable price level at the market of roses in Germany. In order to simulate this current situation, we included this combined attribute in our conjoint experiment.

*2.4. Realization of the Survey and Statistical Analysis*

Data for this online survey were collected by a service provider with an online panel in December 2017. In order to get a representative sample for Germany, 2442 responses were collected in total. Approximately 34.0% of the original respondents were disqualified due to completing the survey in only half of the time (5.5 min) than the average or by showing no variance in regards of their answers. The target group of this survey included people who have bought cut flowers in the last 12 months. The quotation of gender and age followed the average German population structure based on statistical records [57], considering that the most important discounters in Germany (i.e., ALDI and Lidl) showed a very similar age distribution of their customers [58]. Within two weeks of data collection, 1332 participants completed the CBC, and after cleaning this data a total of 1201 responses were retained. Informed consent was obtained from all individual participants included in the study. All procedures performed in this study were in accordance with ethical standards.

The questionnaire and CBC experiment were conducted using Lighthouse Studio version 9.5.3 by Sawtooth Software Inc. The CBC experiment included 10 random choice tasks with three alternative product concepts for each task in addition to a None-option. Introducing the CBC experiment it was indicated that without having any special reason the participant wants to buy a mono bunch of cut roses (10 pieces) and should decide which one of the offered alternatives he/she prefers over the others. It was mentioned in the introduction that the roses were available in the color the participant wished and all of the roses had the same shape and freshness. Based on the principle of orthogonality, a total of 300 questionnaire versions were generated from an overall choice design by the software´s algorithms. The random task generation method 'Balanced Overlap' was applied, which enabled level overlap within individual tasks. By allowing some levels to repeat, and thereby limiting the respondents´ options, the precision of interaction effects between the attributes may improve. The collected data was analyzed with the help of the Hierarchical Bayes algorithm and subsequently a Latent Class model. In the case that heterogeneous consumer preferences might exist, Hierarchical Bayes is the preferred statistical estimation method.

2.4.1. Hierarchical Bayes Estimation

The Hierarchical Bayes procedure was used to estimate the importance values of the four analyzed product attributes as well as the part-worth utility values of the different attribute levels. According to Netzer et al. [59], Hierarchical Bayes is a likelihood-based and random-effects method and consists of two levels. At the upper or population level, it is assumed that individuals' part-worths are described by a multivariate normal distribution. At the lower or individual level, it is assumed that the probability that a respondent will choose a particular alternative is governed by a multinomial logit model [60]. The two levels allow the algorithm to "borrow" missing information about the individual level from the population level. By doing this, the procedure deals with preference heterogeneity by estimating individual level parameters [61]. Hierarchical Bayes analysis uses the average root likelihood (RLH) as a measure of fit for the consistency of the choices of the survey participants [62] and provides the geometric mean of the predicted probabilities. The chance of the alternatives to be chosen in the given conjoint analysis including the None-option was 0.25. Thus, the RLH of our model is 0.65, thus showing a figure which is 2.6 times higher than average chance probability.

2.4.2. Latent Class Analysis

Additionally, a latent class analysis was performed using Lighthouse Studio ver. 9.5.3 to identify consumer segments, each with widely homogeneous preference structures. Instead of finding average part worth utilities for all respondents together, the method estimates the relative probability of each respondent belonging to each segment according to Sawtooth Software [63]. The probabilistic model of the latent class approach applies the distribution of the data, in lieu of clustering with arbitrary chosen distance measures. These probabilities facilitate the re-estimation of the logit weights for

each group to accumulate the overall log-likelihood. As the improvement of the log-likelihood falls under a predefined convergence limit (0.01) the computation terminates. Thereby the whole sample is divided into subgroups with comparable part worth utilities and a discrete assumption of respondents´ heterogeneity. Finally, the number of segments is determined by performance indicators such as the Consistent Akaike Information Criterion (CAIC) or Chi-Square [63].

## 3. Results

### 3.1. Sample Description

Table 2 shows the socio-demographic sample characteristics as well as the purchasing behavior of respondents. The distribution of age and gender follows the structure of the German population [57]. Further, almost all of the respondents (94.7%) state to have purchased in supermarkets or discounters in the recent three months. This, in turn, is important because 80.0% of German consumers at least occasionally purchase cut flowers while shopping for groceries in supermarkets [64]. Half of the participants buy flowers at least once a month and approximately 20.0% even more often (Table 2).

**Table 2.** Socio-demographic structure and purchasing behavior of the respondents.

| Sample Characteristics | Respondents (n = 1201) | Proportion (%) | German Census (2011) (%) |
|---|---|---|---|
| **Gender** | | | |
| Female | 634 | 52.8 | 51.0 |
| Male | 567 | 47.2 | 49.0 |
| **Age** | | | |
| <20 | 78 | 6.5 | 6.9 |
| 20–29 | 159 | 13.2 | 13.8 |
| 30–39 | 169 | 14.1 | 13.9 |
| 40–49 | 195 | 16.2 | 16.8 |
| 50–59 | 228 | 19.0 | 18.0 |
| 60–69 | 164 | 13.7 | 12.9 |
| 70+ | 208 | 17.3 | 17.8 |
| **Purchased in supermarket/ discounter in recent three months** | | | |
| Yes | 1137 | 94.7 | |
| No | 64 | 5.3 | |
| **Purchase frequency of cut flowers** | | | |
| <Once a year | 15 | 1.2 | |
| At least once a year | 343 | 28.6 | |
| At least once a month | 601 | 50.0 | |
| Weekly | 238 | 19.8 | |
| More than once a week | 4 | 0.3 | |
| **Total household net-income (€) per month** | | | |
| <1000 | 90 | 7.5 | |
| 1000–1999 | 271 | 22.6 | |
| 2000–2999 | 319 | 26.6 | |
| 3000–3999 | 217 | 18.1 | |
| 4000+ | 178 | 14.8 | |
| Not answered | 126 | 10.5 | |

### 3.2. Consumer Preferences for Sustainability Labeled Roses

In the first step, the general importance of every attribute and the preferences for individual attribute levels were analyzed. The importance of each attribute was calculated by dividing the

difference between the highest and lowest utility value of each attribute by the sum of all the attributes' differences. The results for the entire sample, which are based on Hierarchical Bayes estimation, are shown in Table 3 highlighting the part-worth utility coefficients of each attribute level and the overall importance of the attributes. The values of the part-worth utilities have been re-scaled to zero-centered differences by the program for better comparability.

**Table 3.** Average utility values (Zero-Centered Diffs) and average importance of attributes of cut roses.

| Attribute Level | Utility Value | SD | Lower 95% CI | Upper 95% CI |
|---|---|---|---|---|
| Big uniform blossoms | 12.0 | 15.1 | 11.1 | 12.8 |
| Small uniform blossoms | −8.8 | 13.7 | −9.6 | −8.0 |
| Non-uniform blossoms | −3.2 | 12.1 | −3.9 | −2.5 |
| Strong scent | 41.8 | 43.3 | 39.4 | 44.3 |
| Weak scent | 4.6 | 18.3 | 3.6 | 5.7 |
| No scent | −46.5 | 33.7 | −48.4 | −44.5 |
| Plastic packaging | −39.8 | 35.5 | −41.8 | −37.8 |
| Paper packaging | 21.9 | 17.1 | 21.0 | 22.9 |
| No packaging | 17.9 | 22.1 | 16.6 | 19.1 |
| Fairtrade/Kenya/moderate price | 74.9 | 55.8 | 71.7 | 78.0 |
| Organic (BIO)/Germany/high price | −49.0 | 78.2 | −53.4 | −44.6 |
| No certification/no COO/low price | −25.9 | 74.7 | −30.1 | −21.7 |
| None-option | −45.1 | 104.2 | −51.0 | −39.2 |
| **Attribute** | **Importance (%)** | **SD** | **Lower 95% CI** | **Upper 95% CI** |
| Blossom size | 7.7 | 5.5 | 7.4 | 8.0 |
| Scent | 25.3 | 15.3 | 24.4 | 26.2 |
| Packaging | 18.1 | 11.5 | 17.5 | 18.8 |
| Labeling/COO/price | 48.9 | 16.9 | 48.0 | 49.9 |

With an average contribution of 48.9% to the overall utility, the attribute "labeling/COO/price" was the most important product attribute of the analyzed roses. Respondents clearly preferred the Fairtrade label with a high positive part-worth utility value. Although both utility values of the other two labeling alternatives were negative, it is remarkable that the no label alternative got a better estimation than organic roses. The second most important attribute was the scent of the roses with an average contribution of 25.3%. Roses with a strong scent were favored over those with weak scent while roses with no scent were negatively evaluated. Concerning the type of packaging which had an overall contribution of 18.1% to the total utility of respondents, there was no clear differentiation in preferences for paper packaging or no packaging of the roses. However, respondents clearly rejected plastic packaging of the roses. Finally, size of the blossom of the roses was the least important attribute, with a remaining average contribution of around 7.7%. Thereby big blossom sizes were preferred over small or varying sizes. The None-option produced a clear loss of benefit, which indicates that consumers have little interest in choosing no product at all. However, the standard deviations of the part worth utilities as well as the importance figures in Table 3 show rather high figures, what can be interpreted as a hint for a potential heterogeneity in the sample data. Thus, a segmentation analysis was applied using a Latent class model in a subsequent step.

*3.3. Segmentation of the Respondents*

Different market segments were identified through a latent class analysis (LCA) which indicated varying consumer preferences. Solutions with one to up to five groups were calculated to find out the best fit for our data. Finally, the three-group solution was chosen respecting indicators according to Sawtooth Software [63] such as the Log-likelihood, Percent Certainty, CAIC and Relative Chi-Square which are documented in Table 4. Although these values do not have an optimum for a solution with three clusters, decreasing distances indicate a meaningful solution. The percent certainty value as



an index for goodness of the solution compared to the null solution, shows that we improved the data with approximately 28.0% for the three-group version (Table 4), as well as an average maximum membership probability of 93.0% suggests a very good fit of the model. Furthermore, solutions with more clusters produce very small segments thereby complicating the interpretation of the results. Whereas, the three-group model provides a balanced and interpretable distribution of the groups' membership ranging between 30.2% and 35.5% of the sample as well as the results are stable and reproducible as repeated random tests showed [60].

**Table 4.** Goodness of fit of the subgroup estimation for the LCA.

| Number of Groups | Log-Likelihood | Pct Cert | CAIC | Chi-Square | Relative Chi-Square |
|---|---|---|---|---|---|
| 1 | −13,751.47 | 17.41 | 27,596.48 | 5795.85 | 643.98 |
| 2 | −12,726.71 | 23.56 | 25,650.89 | 7845.37 | 412.91 |
| 3 | −11,972.43 | 28.09 | 24,246.26 | 9353.94 | 322.55 |
| 4 | −11,492.46 | 30.97 | 23,390.27 | 10,313.87 | 264.46 |
| 5 | −11,162.03 | 32.96 | 22,833.35 | 10,974.73 | 223.97 |

Segment sizes, individual utilities for the attribute levels and the particular importance of attributes for the three-group solution are listed in Table 5. As all t ratios exceed, by far, the threshold of 1.96, the estimated coefficients are significantly different from 0 at the 99.0% confidence level. Additionally, the very small standard errors ranging from 0.03 to 0.07 also indicate that the examined attributes significantly contribute to the group membership assignment. In the following, the three identified groups are characterized according to the results of the latent class model.

**Table 5.** Average utility values and importance of attributes (zero-centered differences) per group.

| | Segment Size | | |
|---|---|---|---|
| | **Group 1** | **Group 2** | **Group 3** |
| | **34.3%** | **30.2%** | **35.5%** |
| **Attribute level** | Utility value per group | | |
| Big uniform blossoms | 15.06 | 19.63 | 7.08 |
| Small uniform blossoms | −14.33 | −11.22 | −6.14 |
| Non-uniform blossoms | −0.73 | −8.42 | −0.94 |
| Strong scent | 69.68 | 62.86 | 22.91 |
| Weak scent | −2.89 | −4.78 | 11.29 |
| No scent | −66.79 | −58.08 | −34.20 |
| Plastic | −61.42 | −14.68 | −54.13 |
| Paper | 31.57 | 11.45 | 29.09 |
| No packaging | 29.85 | 3.24 | 25.04 |
| Fairtrade/Kenya/€1.99 | 32.98 | 57.37 | 155.48 |
| Organic (BIO)/GER/€3.99 | 54.08 | −139.73 | −90.98 |
| No certification/no COO/€1.65 | −87.06 | 82.36 | −64.50 |
| None-option | −110.06 | 42.08 | −84.34 |
| **Attribute** | Importance per group (%) | | |
| Blossom size | 7.35 | 7.71 | 3.31 |
| Scent | 34.12 | 30.23 | 14.28 |
| Packaging | 23.25 | 6.53 | 20.80 |
| Labeling /COO/price | 35.29 | 55.52 | 61.61 |

### 3.3.1. Group 1—Organic Rose Enthusiasts

According to the results summarized in Table 5, one characteristic feature for Group 1 is the almost equal importance of the combined attribute labeling/COO/price (35.3%) and the attribute scent (34.1%). Additionally, this group shows the highest relevance for packaging (23.3%) compared to the

other two segments. Despite the highest price level, members of Group 1 have a stronger preference for organic roses produced in Germany compared to Fairtrade roses from Kenya and clearly reject the cheapest roses with no labeling. As the attribute scent is quite important for them, and roses with a strong scent are highly preferred over the other alternatives. Members value environmentally friendly packaging and prefer paper and no packaging over the plastic alternative. A very low utility for the None-option (−110.1) suggests that members of Group 1 prioritize in particular one of their preferred product alternatives instead of choosing no product in the CBC experiment. Due to their high preference for organically produced cut roses, Group 1 is described as "Organic rose enthusiasts."

### 3.3.2. Group 2—Price Hunters

Regarding the relevance of the analyzed attributes, Group 2 shows a higher relevance to the combined attribute labeling/COO/price (55.5%) compared to the first group while the importance of scent (30.2%) is in the same range. One specific characteristic of this group is the very low relevance that is given to the packaging of roses (6.5%). Investigating the part worth utilities of the attribute levels shows a strong rejection of organic roses produced in Germany at the highest price level (€3.99/10 pcs.) with a part worth value of −139.7 among the members of the second group. In contrast to the two other segments, the no label roses with the lowest price (€1.65/10 pcs.) score best in Group 2. With respect to blossom size and scent the representatives of this group express similar preferences as Group 1. Concerning packaging we note a very low importance for this attribute with still the paper packaging option being preferred over plastic packaging. Another specific characteristic of this group is the positive part worth figure of 42.1 for the None-option indicating that the members often decided to choose none of the presented product alternatives during the CBC experiment. Due to their positive evaluation of the low-priced roses without any label, we call this group "price hunters."

### 3.3.3. Group 3—Fairtrade Advocates

As Table 5 indicates, the third group gives the highest relevance to the attribute labeling/COO/price (61.6%) in comparison to the other two groups. Packaging is following in the importance list with a value of 20.8%, and thus being almost equally important as for Group 1. The members of Group 2 give the scent (14.3%) and the blossom size (3.3%) of the roses half of the importance compared to the other two segments. When analyzing the part worth utilities of the different attribute levels, this group presents the highest score of all groups for Fairtrade roses of Kenya at medium prices (155.5) and clearly rejects organic roses produced in Germany offered for high prices (−90.98) as well as the roses without label and low prices (−64.5). In addition, environmentally friendly packaging, especially paper (29.1) is relevant for the members of Group 3 and they strongly reject plastic packaging (−54.1) on a comparable level as Group 1. In accordance to the negative part worth value for the None-option (−84.3) the members of Group 3 appear to have a clear idea what they are looking for when choosing roses. Thus, they opted for one of their preferred product alternatives instead of choosing nothing in the CBC experiment. As the results of the part worth values for the different attribute levels of scent and blossom show the same tendencies in Group 3 as in the other two groups, we emphasize the high preference for Fairtrade roses and name this group "Fairtrade advocates."

### 3.4. Socio-Economic Characteristics of the Groups

The socio-economic characteristics consisting of gender, age, net household income per year, purchase frequency and yearly expenditures on flowers of the three groups are presented in Table 6. After performing *t*-tests, a few statistically significant (*p* = 0.05) differences between the groups can be observed. These differences support the results of Table 5 and emphasize the three-group classification of the LCA. The mean values of gender distribution show a balanced composition between men and women of the Price hunter group whereas the other two groups have a higher proportion of female respondents. Fairtrade advocates seem to be the oldest group with 21.0% of them even being older than 70 years and the majority between 40 and 69 years. The age distribution for Price hunters and for

Organic rose enthusiasts indicate a peak between 20 and 59 years. Moreover, the latter group has the highest proportion of consumers younger than 20 years of all three groups and thereby members of the Fairtrade advocates are significantly older. In term of net household income, Organic rose enthusiasts seem to have in tendency higher income figures compared to the other two groups. Regarding yearly expenditures on flowers and purchase frequency, Price hunters spend significantly lower amounts for flowers per year and purchase them less frequently compared to the other two groups (Table 6).

**Table 6.** Socio-economic characteristics of the three groups based on the LCA segmentation (in %).

| | | Organic Rose Enthusiasts (Group 1) | Price Hunters (Group 2) | Fairtrade Advocates (Group 3) |
|---|---|---|---|---|
| | **n** | **364** | **408** | **429** |
| **Gender** | female | 54.7 | 49.5 | 53.8 |
| | male | 45.3 | 50.5 | 46.2 |
| **Age** | <20 | 8.6 | 4.9 | 5.8 [c] |
| | 20–29 | 13.7 [c] | 14.3 | 11.9 [c] |
| | 30–39 | 15.0 [c] | 15.4 | 12.1 [c] |
| | 40–49 | 16.9 [c] | 16.5 | 15.4 [c] |
| | 50–59 | 18.6 [c] | 20.3 | 18.2 [c] |
| | 60–69 | 13.0 [c] | 12.1 | 15.6 [c] |
| | 70+ | 14.2 [c] | 16.5 | 21.0 [c] |
| **Net household income/month** | <€1000 | 8.6 | 7.4 | 6.5 |
| | €1000–€1999 | 18.9 | 26.4 | 22.8 |
| | €2000–€2999 | 27.5 | 25.8 | 26.3 |
| | €3000–€3999 | 16.2 | 17.6 | 20.3 |
| | €4000+ | 18.1 | 11.5 | 14.5 |
| | n.a. | 10.8 | 11.3 | 9.6 |
| **Purchase frequency of flowers** | <once a year | 1.2 [a] | 1.4 [a,b] | 1.2 [b] |
| | more than once a year | 27.0 [a] | 31.9 [a,b] | 27.3 [b] |
| | once a month | 49.5 [a] | 51.4 [a,b] | 49.4 [b] |
| | weekly | 22.1 [a] | 15.1 [a,b] | 21.7 [b] |
| | multiple times a week | 0.2 [a] | 0.3 [a,b] | 0.5 [b] |
| **Expenditure on flowers/year** | <€50 | 21.3 [a] | 31.9 [a,b] | 21.9 [b] |
| | €50–€99 | 27.2 [a] | 29.4 [a,b] | 29.4 [b] |
| | €100–€299 | 35.3 [a] | 29.7 [a,b] | 34.0 [b] |
| | €300–€499 | 11.5 [a] | 6.3 [a,b] | 11.2 [b] |
| | €500+ | 2.9 [a] | 1.1 [a,b] | 2.3 [b] |
| | n.a. | 1.7 [a] | 1.6 [a,b] | 1.2 [b] |

Significant difference on a 0.05 level between Groups [a] 1 and 2, [b] 2 and 3, [c] 1 and 3.

## 4. Discussion

Based on the consumers´ utilities, preferences, and the emerging latent groups, this study reveals that the majority of German consumers have general interest in sustainability labeled horticultural products. The attribute "labeling/COO/price" consisting of the combined components of a labeling certificate, the country of origin of the plant, and the price was considered being the most important one in our study. These findings are supported by previous studies analyzing at least single dimensions of sustainability issues in Germany [21,22] and other countries [23,33,34,53]. In the study on hand, the used packaging material was also regarded as relevant for consumer preference evaluation of the analyzed cut roses, what is also supported by previous studies [50–52,65]. The inherent characteristics of the roses contributed to around one third to the overall utility of respondents with scent of the roses being the dominant attribute which is partially supported by previous studies [48].

The results of the latent class analysis give clear indications of the existence of consumer heterogeneity with major differences between the three groups being identified with regard to the respondents' part-worth utility values as well as the importance of the analyzed product attributes.

Within this study, two groups were clearly in favor of sustainability labeled cut roses being aware and sensitive either to the social dimension of sustainability represented by the Fairtrade label or they strongly preferred organic roses produced in Germany. Based on the loss of benefit for the None-option in these groups, the preferred attributes seem to convince the members for the sustainably produced alternatives. The linked higher product price was accepted by the respondents of these two segments. Altogether, respondents belonging to these two groups made up 69.8% of the sample. In contrast, around 30.2% of the respondents were only partially interested in sustainability labeled cut roses. These respondents seemed to take more notice of low prices (which were connected to the No-label alternative) and did not expect specific sustainability attributes when purchasing roses. The members of this group were much likely to refrain from purchasing any other alternative than their preferred no-label product. As this study examined cut roses as an example for horticultural products, the results probably cannot be transferred to all horticultural products in Germany.

Within the "labeling/COO/price" attribute, Fairtrade roses were strongly preferred by the entire sample. This was particularly true for the group of the Fairtrade advocates, but Fairtrade roses received positive part-worth utility values in all three identified segments (Table 5). This finding might be due to the fact that a rather moderate price (€1.99/10 pcs.) was linked to the Fairtrade certificate and that shoppers in discounters and supermarkets in Germany might be rather familiar with this label due to the growing market volume of fair-traded plants and other products in recent years [66]. The strong preferences for Fairtrade roses did not seem to be restricted by the fact that Kenya was mentioned as being the country of origin in the study on hand although previous studies on horticultural products found preferences for domestically grown plants in Germany [21,22,32] and other countries [33,34]. A similar tendency for consumer preferences for domestically or regionally produced products can be observed in the food markets as well (e.g., [67–69]).

In contrast to the Fairtrade attribute level, the organic label of the analyzed cut roses only got positive consumer evaluations in the group of the Organic rose enthusiasts and was clearly rejected in the other two clusters as well as within the entire sample. This result might be due to the fact that a price of €3.99 for a bunch of 10 roses was linked to the used Bioland certificate which might be perceived by shoppers in discounters and supermarkets in Germany as being too high for cut roses. In general, the high consumer acceptance of organic labels which can be observed for the food market in Germany [7,69] does not seem to correspond with analogous consumers' preferences for non-edible organic products. As some studies related to the food sector showed that consumers' preference for an organic label mainly depend on their perceived positive health aspects [24,25,70], it can be assumed that health aspects are being less important related to non-edible products like cut roses thus leading to lower preferences of the Bioland labelled roses. Additionally, Loureiro and Lotade [71] argued that consumers were not quite aware or could barely imagine the environmental impact of coffee cultivations in distant regions, what might be also transferable to the analyzed cut roses.

The results concerning the levels of the attribute packaging were homogeneous over all three clusters as well as for the entire sample with paper packaging or no packaging of the cut roses being preferred over plastic packaging. This result might be influenced by the current public debate on reducing plastic waste in Germany which popped up during the realization of the survey. However, previous studies for other products also showed a positive consumer evaluation in particular for paper packaging [50,51]. Respondents' preferences related to the analyzed levels of the attributes scent and blossom size of the cut roses showed the expected tendencies. Strong scent is preferred over weak or no scent of the roses [72] as well as uniform big blossom sizes being positively evaluated by all three clusters as well as the entire sample, in accordance with the findings of Prince et al. [73].

A choice-based conjoint (CBC) analysis was conducted to analyze consumers' preferences of sustainability labeled roses in Germany. CBC is regarded as a reasonable instrument to measure consumer preferences and thus has practical relevance by simulating real life purchase situations [74]. Concerning sustainability oriented attitude and behavior, respondents often exaggerate their positive behavior patterns in a conventional interview situation. This fundamental methodological problem

of surveys or interviews can be alleviated by a choice-based conjoint analysis. Within such a study, respondents tend to focus on particular characteristics in a choice set and to seek strategies which simplify their decision. This is generally regarded not as a limitation of the study results as this selection strategy corresponds to the real life behavior of consumers [75,76]. A None-option was used, in which respondents had to choose between the different sustainability labeled roses and a conventional alternative. In this way, the real market situation was simulated with a restricted set of product alternatives and respondents who refused the sustainability labeled roses could be identified. Furthermore, a latent class analysis identified three consumer groups with different preferences based on the respondents' utility functions and gave additional detailed insight into consumer heterogeneity in the analyzed market.

Nonetheless, the methodological approach of our study faces some limitations. Although a real purchase situation was simulated, the buying process was hypothetical in the sense that respondents did not have to spend any money [75,77]. An investigation at the point of sale with a very limited number of defined product alternatives would be a promising approach in future studies but could not be realized in the study on hand due to practical barriers like e.g., availability of the respective plants over a longer period of time or willingness of retail chains to participate in such an experiment. Future studies could examine further examples of sustainability labeled products and possible target groups. Another limitation of this study represents the fact that this survey was restricted to respondents living in Germany. Although the German market for horticultural plants in general and cut flowers in particular has the highest market volume in Europe [12], results from other countries could enhance the knowledge about consumers' interests in sustainability labeled horticultural plants, which is a topic of worldwide relevance. Finally, the use of an online panel can be regarded as a further limitation. Thereby respondents across Germany could easily be reached and—based on the quota according to age and gender—the relevant target groups were representative in our sample, but members of online panels are often more familiar with standardized surveys what could have an influence on their responses [78]. An attempt to minimize this effect was made by excluding respondents with uniform response patterns and very short answering times.

## 5. Conclusions

The most important aim of the study on hand is to identify sustainability criteria, which support consumer choices related to the analyzed cut roses. Thereby we could show that, in particular, social aspects (e.g., reduction of child labor, improving working conditions and income of workers in low-income countries) are of specific relevance in this area that are often connected with the Fairtrade label. Additionally, the inherent quality of the products (in our study represented by the blossom and scent of the cut roses) plays an important role in the product estimation of the consumers—an aspect that was often not considered in previous studies. Furthermore, we could demonstrate the existence of consumer heterogeneity not only within the entire sample but also related to the different dimensions of sustainability. Thereby, almost all consumers positively assessed the social and "fair" dimension of the analyzed cut roses while only around one third of them are in favor of the organic roses that address the ecological dimension of sustainability in particular.

Altogether, our study provides clear insight into relevant sustainability criteria for consumers related to sustainability labeled roses. Properties enhancing social and environmental sustainability are an option for producers to append additional value to horticultural products, which often suffer under price reduction tendencies. The study shows that there are differing target groups in this market with varying preferences and price expectations which can be targeted by producers and retailers. Some of these target groups even accept a (limited) price premium for sustainable product variants. Producers and distributors are recommended to fulfil the specific expectations of the different groups identified in this study.

**Author Contributions:** K.M. was responsible for supervision of the experimental design project administration, editing the written manuscript and funding acquisition. Field work, data collection and statistical analysis,

literature review, phrasing and finalization of original draft and final manuscript were carried out by D.B.-K. Conceptualization and methodology were collectively elaborated.

**Funding:** This research was funded by the Bavarian Ministry of Education and Culture, Science and Arts, grant number VIII.2-F1116.WE/26/28.

**Acknowledgments:** The authors appreciate financial support of the study by the Bavarian Ministry of Education and Culture, Science and Arts but are solely responsible for the content of the manuscript.

**Conflicts of Interest:** The authors declare no conflict of interest. The funders had no role in the design of the study; in the collection, analyses, or interpretation of data; in the writing of the manuscript, or in the decision to publish the results.

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
