# Peer review of "Consumer Preferences of Sustainability Labeled Cut Roses in Germany"

_sustainability, doi:10.3390/su11123358_

Round 1
Reviewer 1 Report
The paper deals with a relevant topic regarding the importance of sustainable attribute in Cut rose. The paper has a clear objective with a good review of most important paper that focused on sustainable label. In all cases, it is not clear the additional contribution of this paper compared to literature. Is the method used is never applied before in Rose preferences?. Is the sustainable label rose?. Please give more insight.
What I miss in the introduction some comments regarding the implication of such label at farm level mainly con production cost. This information will be valuable to understand if the
Line 45: define AMI
I was confused at the beginning of this section. The authors started to talk about Conjoint analysis and later mentioned the Discrete Choice Experiment as the same methods applied. This part should be clarified since this can bring readers to confusion. Please have a look on this paper.
Louviere, J. J., Flynn, T. N., & Carson, R. T. (2010). Discrete choice experiments are not conjoint analysis. Journal of Choice Modelling, 3(3), 57-72.
Figure 1 is highly relevant since it help authors to clearly define which attribute to introduce in the final study. Did the authors ensure the presence of the three dimension of the sustainability concept? (social, environmental and economic): I realized that the economic aspect is underrepresented. Is this related to the expert opinions and literature review?
In all cases, it is not clear how the attribute selection presented in Table 1 was related to the aspects that consumers take into consideration when purchase rose (from Figure 1)
My main concern is related to you experimental design that is not well defined. In particular the Last attribute Labeling /COO/price is very strange and I never see this way to link three different attributes in one fixed set and fix the levels. In this case, there is no any trade-off because always the support of developing countries will be only related with €1.99; the support of regional production only related with the €3.99 and no brand with €1.65. The price level is always constant and only the other attributes can vary?. This is a big limitation to your design as it is not clear at all which experimental design you followed (orthogonal, fractional factorial, D optimal, …)
Why only focus on the relative importance of the attribute and not on the WTP of sustainable label? Can you attach all choice sets so I can understand your design? What do you mean by 300 questionnaire version? You used 300 different choice design?¡¡
The same for the sentence forcing the respondents to decide between the same attribute levels. Previously you mentioned you included the no choice option. So it is not forced choice. Again he design is confusing.
More details on the modeling and results of the latent class are needed. The socio-economic profile of the different segment is not reported.
The loss of benefit of the non-choice option is not necessary an indicator to preference for sustainable label. It is not how you get this conclusion from your results. The reluctance to the No-choice could reveal preference for any combination of your attribute and not only the sustainable one.
Summarizing, I am very concerned with your design and its capacity to correctly estimate the utilities. Last attribute represent 3 attributes that are always have constant combination. Thus, which trade-off the consumers are going to make if the price is always constant? This point should be clarified.
Author Response
Dear Reviewer,
thank you very much for your time and effort you have spent on reviewing our manuscript, which was relegated for major revision until 25 May (deadline extended to 28 May, granted by assigned editor). We have read all your comments and worked hard on addressing all the remarks, and finally we followed almost all of the reviewer’s requests. Please find all our responses to your comments below; the line numbers are based on the PDF document with the title “503053-Revised_ver2_CLEAN”. We have also added an additional word file where all our changes are tracked “503053-Revised_ver2_TRACK”, so you have an overview of all the changes applied.
Thank you in advance for supporting us with your expertise.
Comments and Suggestions for Authors
The paper deals with a relevant topic regarding the importance of sustainable attribute in Cut rose. The paper has a clear objective with a good review of most important paper that focused on sustainable label. In all cases, it is not clear the additional contribution of this paper compared to literature. Is the method used is never applied before in Rose preferences?. Is the sustainable label rose?. Please give more insight.
Lines 95-107 and 552-572: Please find our modifications at the end of the introduction and also in a new conclusion section at the end of the manuscript. Since another reviewer also suggested us to work on this part, we really worked on explaining our contribution in a meaningful manner. We see the main contribution in the following fields:
Our study analyses the relevance and estimations related to different dimensions of sustainability in the view of consumers, include the inherent quality of the analyzed product (i.e. cut roses) in this process and aims to identify consumer heterogeneity not only within the entire target group but also related to the different dimensions of sustainability.
What I miss in the introduction some comments regarding the implication of such label at farm level mainly con production cost. This information will be valuable to understand if the
In the lines between 62-64 you find some small changes regarding your comment, which hopefully is in line with what you were expecting.
Line 45: define AMI
AMI=agricultural market information company; please find the adjustment in the manuscript with all the other tracked changes.
I was confused at the beginning of this section. The authors started to talk about Conjoint analysis and later mentioned the Discrete Choice Experiment as the same methods applied. This part should be clarified since this can bring readers to confusion. Please have a look on this paper.
Louviere, J. J., Flynn, T. N., & Carson, R. T. (2010). Discrete choice experiments are not conjoint analysis. Journal of Choice Modelling, 3(3), 57-72.
We are genuinely thankful for you pointing out this confusion and for adding the explanation in form of this comprehensible paper. We hope to fulfill all the reviewers´ requirements by shortening this part, since the length and elaborateness of this particular section was criticized by the second reviewer. Further, we made some minor changes in the manuscript (e.g. abstract – line 12) to avoid misunderstandings.
Changes in Lines 113-124: We have deleted some incoherent parts, which were misleading and included your suggested literature.
Figure 1 is highly relevant since it help authors to clearly define which attribute to introduce in the final study. Did the authors ensure the presence of the three dimension of the sustainability concept? (social, environmental and economic): I realized that the economic aspect is underrepresented. Is this related to the expert opinions and literature review?
Thank you for this remark! We mainly focused on environmental and social aspects according to the findings of the expert interviews. In contrary to the views of representatives from production companies, in the view of private consumers the economic aspect is less relevant (in particular the economic performance of retail or production companies). Therefore we only included price as (“economic”) variable that may have a direct effect on consumers’ choices.
Under lines 159-160 we added a short reasoning, why we did not focus on the economic part.
In all cases, it is not clear how the attribute selection presented in Table 1 was related to the aspects that consumers take into consideration when purchase rose (from Figure 1)
We asked consumers to rank the highly relevant attributes mentioned by the experts in the pre-study (results are presented in Figure 1), compared the results of the pre-study with previous literature and combined it with potentially influencing factors at a real purchase situation to extract the most characteristic attributes. Fairtrade mainly represents the social aspect, the Eco label the ecological aspects of production and distribution of cut roses. Packaging was mentioned as important factor in the consumer survey of the pre-study. Blossom size and scent are indicators of the inherent quality of the analyzed cut roses. We refrained to add the color as a separate attribute, because this in a very individual attribute in our point of view.
Line 204-215: Please find our extended argumentation on how we included the attributes.
My main concern is related to you experimental design that is not well defined. In particular the Last attribute Labeling /COO/price is very strange and I never see this way to link three different attributes in one fixed set and fix the levels. In this case, there is no any trade-off because always the support of developing countries will be only related with €1.99; the support of regional production only related with the €3.99 and no brand with €1.65. The price level is always constant and only the other attributes can vary?. This is a big limitation to your design as it is not clear at all which experimental design you followed (orthogonal, fractional factorial, D optimal, …)
The experimental design followed the orthogonal principle; the attributes were distributed evenly balanced and the levels were treated independently.
We understand and are thankful for your concern regarding the limitation, although with all due respect, we deliberately chose to link the three components (label/COO/price) based on the following considerations: 1) We intended to consider the current market and price situation for cut roses (sold in supermarkets) in our choice experiment. Thereby the observed market prices in Germany show the tendencies we applied. Roses with no labels are always cheaper than the other alternatives. Fairtrade roses - referring to empirical checks of regional stores and literature insights - always show a somewhat higher price than the no-label version (therefore middle-ranged price in our experiment). On the other hand Eco-roses (as a niche product), produced in Germany are sold at the highest average price compared to the other two alternatives. 2) Eco-roses are very rare to be found on the German market and stand out because of their special region of origin: the production in Germany is more expensive (mainly due to energy costs), therefore the highest price for this product is reasonable. 3) Most of the Fairtrade roses are produced in Kenia and fewer in Ecuador, but it is impossible that they are produced in Germany. Therefore we decided to link the country with the highest numbers (Kenia) with the label, since it only would have distorted our results, if we would have offered the participants a non-existing alternative. 4) Based on our findings in the stores, the non-labeled roses also do not display their country of origin. Hence we tried to simulate a purchase situation as real as possible. Altogether, our point is that it is not realistic to include e.g. eco-roses from Kenia, Fairtrade-roses from Germany, or eco-roses at the cheapest price, as well as none-labeled roses as the most expensive alternative, because participants will not find these products in German stores. Therefore we decided to link the three components label/COO/price and present realistic options in the choice experiment.
Why only focus on the relative importance of the attribute and not on the WTP of sustainable label? Can you attach all choice sets so I can understand your design? What do you mean by 300 questionnaire version? You used 300 different choice design?¡¡
We focused on the relative importance of the attributes and the part-worth utility values of the attribute levels since we did not intend to carry out another Willingness-to pay study. Instead our main target was to identify those sustainability criteria, which support consumer choices related to the analyzed cut roses.
From a single overall choice design the software´s algorithms created by default 300 unique questionnaire versions, all of them according to the principle of orthogonality (balanced and independent distribution of questions). As the participants are randomly selected to receive one particular questionnaire version, approximately only four of the participants (from app. N=1200) received the same version with 10 random and two fixed profiles with the profiles being in the same order. This procedure reduces order bias when carrying out the choice experiment.
Between lines 294-307 you will find some adjustments regarding your previous annotation to our CBC model; also please find the choice-sets attached as a separate PDF-file.
The same for the sentence forcing the respondents to decide between the same attribute levels. Previously you mentioned you included the no choice option. So it is not forced choice. Again he design is confusing.
Thank you for pointing this out; the wording here in this case was indeed misleading. Please find our adjustment in the lines 303-305.
More details on the modeling and results of the latent class are needed. The socio-economic profile of the different segment is not reported.
Following your advice, we added the socio-economic characteristics of the different groups. Originally it was left out to reduce the page count of the paper and because the differences between the groups only display minor significant effects.
For listing the relevant results of our LCA we applied advices of the original manuscripts of the Software used for computation. We reported the most important performance indicators (like Log-likelihood, CAIC, Chi-Square, Percent certainty, membership probability, significant t-ratios, standard errors, of course utility values and importances) in the manuscript. Would you be so kind to specify which exact details you feel that are lacking?
The loss of benefit of the non-choice option is not necessary an indicator to preference for sustainable label. It is not how you get this conclusion from your results. The reluctance to the No-choice could reveal preference for any combination of your attribute and not only the sustainable one.
Thank you also for this comment, we agree with your argumentation. Therefore we changed the wording in the manuscript stating now: The None-option produced a clear loss of benefit, which indicates that consumers have little interest in choosing no product at all. (Lines 365-366)
Summarizing, I am very concerned with your design and its capacity to correctly estimate the utilities. Last attribute represent 3 attributes that are always have constant combination. Thus, which trade-off the consumers are going to make if the price is always constant? This point should be clarified.
Thank you again for all your comments and your effort! Please also see our argumentation above regarding this remark. In order to respond to your last question, we aimed to confront the respondents of the choice experiment with a realistic market situation, where these 3 linked attributes cannot be separated. They either choose constantly the alternative, which they prefer over another, are forced to make trade-offs regarding packaging or the intrinsic characteristics of the cut roses, or they refuse to choose any of the alternatives. We argue that especially in combination with the socio-economic characteristics and the differentiation of the LCA groups, the model discloses the utilities and preferences in a reasonable way.

Reviewer 2 Report
In some cases, it is necessary to reference correctly so as not to lose the thread of the speech. For example, see lines 35, 40, 76, 78, ...
There is some typographical error. Eg line 430.
On line 86 you talk about aim of the project? Also on line 504, you talk about a project.
A most profound review of the literature on the subject under study is lacking and, perhaps, it can be summarized a bit the explanation of the method, Choice-based-conjoint analysis, which is widely known.
It is necessary to better justify the importance of this research. What does it contribute? What is it for? This must be done, above all, in the introduction section. It is also convenient to do it in a new section of conclusions where it is pointed to the main conclusion of the paper responding to the objectives (without discussion), by way of theoretical and practical implications.
Author Response
Dear Reviewer,
thank you very much for your time and effort you have spent on reviewing our manuscript, which was relegated for major revision until 25 May (deadline extended to 28 May, granted by assigned editor). We have read all your comments and worked hard on addressing all the remarks, and finally we followed almost all of the reviewer’s requests. Please find all our responses to your comments below; the line numbers are based on the PDF document with the title “503053-Revised_ver2_CLEAN”. We have also added an additional word file where all our changes are tracked “503053-Revised_ver2_TRACK”, so you have an overview of all the changes applied.
Thank you in advance for supporting us with your expertise.
Comments and Suggestions for Authors
In some cases, it is necessary to reference correctly so as not to lose the thread of the speech. For example, see lines 35, 40, 76, 78, ...
There is some typographical error. Eg line 430.
On line 86 you talk about aim of the project? Also on line 504, you talk about a project.
Thank you very much for pointing out these previous inaccuracies and are corrected in the new version of the manuscript!
A most profound review of the literature on the subject under study is lacking and, perhaps, it can be summarized a bit the explanation of the method, Choice-based-conjoint analysis, which is widely known.
We appreciate your suggestions. In order to fulfill the comments of the first reviewer, we shortened the part describing a Choice-based conjoint analysis, and tried to elaborate on it more accurately.
However, we struggled a lot how to resolve the “lack of a profound literature review” which you mentioned above. Since we did not receive any other comment on that by the two other reviewers, would you mind giving us a more elaborate hint, how we could proceed in your favor? Up to now, we were convinced that we have worked out the most important aspects of the topic as e.g. sustainability, sustainable consumption (in the field of flowers), market of ornamental plants and especially roses, sustainability labels in this field, and the role of sustainability labels related to consumers.
It is necessary to better justify the importance of this research. What does it contribute? What is it for? This must be done, above all, in the introduction section. It is also convenient to do it in a new section of conclusions where it is pointed to the main conclusion of the paper responding to the objectives (without discussion), by way of theoretical and practical implications.
Thank you for pointing this out and also for suggesting an example on how to comply. We worked hard on the improvement in the manuscript and also added a separate section for the conclusions.
Lines 95-107 and 552-572: Please find our enhancements at the end of the introduction and also in a new conclusion section at the end of the manuscript. Since another reviewer also suggested us to work on this part, we really put effort on elaborating our contribution in a meaningful manner.
Reviewer 3 Report
The objective of this research is to identify sustainability criteria when purchasing sustainability labeled cut roses in food retailing outlets in Germany.
I find it very interesting. The introduction seemed appropriate to present the problem, Material and methods section is sufficient. The methodology seems correct. The results and discussion are coherent. Authors also point out the limitations of the study. The final conclusion is completely right and shows the social implications of the study.
Congratulations!
Author Response
The objective of this research is to identify sustainability criteria when purchasing sustainability labeled cut roses in food retailing outlets in Germany.
I find it very interesting. The introduction seemed appropriate to present the problem, Material and methods section is sufficient. The methodology seems correct. The results and discussion are coherent. Authors also point out the limitations of the study. The final conclusion is completely right and shows the social implications of the study.
Congratulations!
Dear Reviewer,
We very much appreciate your positive review and want to thank you for your welcoming words. In order to address some changes requested by the other two reviewers although, we had to perform some changes on the original manuscript. Nevertheless, we hope that our modifications also meet your expectations.
Thank you for your effort and support!
Round 2
Reviewer 1 Report
Authors made a good explanation and justification of my main concern related to the design. They justified the inclusion of an attribute always attached to a fixed price. The supplementary material attached gave a comprehensive explanation of the design they used.